# When are ensembles really effective?

**Ryan Theisen**
Department of Statistics
University of California, Berkeley
theisen@berkeley.edu

**Hyunsuk Kim**
Department of Statistics
University of California, Berkeley
hyskim7@berkeley.edu

**Yaoqing Yang**
Department of Computer Science
Dartmouth College
Yaoqing.Yang@dartmouth.edu

**Liam Hodgkinson**
School of Mathematics and Statistics
University of Melbourne, Australia
lhodgkinson@unimelb.edu.au

**Michael W. Mahoney**
International Computer Science Institute
Lawrence Berkeley National Laboratory
and Department of Statistics
University of California, Berkeley
mmahoney@stat.berkeley.edu

## Abstract

Ensembling has a long history in statistical data analysis, with many impactful applications. However, in many modern machine learning settings, the benefits of ensembling are less ubiquitous and less obvious. We study, both theoretically and empirically, the fundamental question of when ensembling yields significant performance improvements in classification tasks. Theoretically, we prove new results relating the *ensemble improvement rate* (a measure of how much ensembling decreases the error rate versus a single model, on a relative scale) to the *disagreement-error ratio*. We show that ensembling improves performance significantly whenever the disagreement rate is large relative to the average error rate; and that, conversely, one classifier is often enough whenever the disagreement rate is low relative to the average error rate. On the way to proving these results, we derive, under a mild condition called *competence*, improved upper and lower bounds on the average test error rate of the majority vote classifier. To complement this theory, we study ensembling empirically in a variety of settings, verifying the predictions made by our theory, and identifying practical scenarios where ensembling does and does not result in large performance improvements. Perhaps most notably, we demonstrate a distinct difference in behavior between interpolating models (popular in current practice) and non-interpolating models (such as tree-based methods, where ensembling is popular), demonstrating that ensembling helps considerably more in the latter case than in the former.

## 1 Introduction

The fundamental ideas underlying ensemble methods can be traced back at least two centuries, with Condorcet's Jury Theorem among its earliest developments [Con85]. This result asserts that if each juror on a jury makes a correct decision independently and with the same probability $p > 1/2$, then the majority decision of the jury is more likely to be correct with each additional juror. The general principle of aggregating knowledge across imperfectly correlated sources is intuitive, and it has motivated many ensemble methods used in modern statistics and machine learning practice. Among

37th Conference on Neural Information Processing Systems (NeurIPS 2023).

these, tree-based methods like random forests [Bre01] and XGBoost [CG16] are some of the most effective and widely-used.

With the growing popularity of deep learning, a number of approaches have been proposed for ensembling neural networks. Perhaps the simplest of them are so-called deep ensembles, which are ensembles of neural networks trained from independent initializations [ABP+22, ABPC22, FHL19]. In some cases, it has been claimed that such deep ensembles provide significant improvement in performance [FHL19, OFR+19, ALMV20]. Such ensembles have also been used to obtain uncertainty estimates for prediction [LPB17] and to provide more robust predictions under distributional shift. However, the benefits of deep ensembling are not universally accepted. Indeed, other works have found that ensembling is less necessary for larger models, and that in some cases a single larger model can perform as well as an ensemble of smaller models [HVD+15, BCNM06, GJS+20, ABP+22]. Similarly mixed results, where empirical performance does not conform with intuitions and popular theoretical expectations, have been reported in the Bayesian approach to deep learning [INLW21]. Furthermore, an often-cited practical issue with ensembling, especially of large neural networks, is the constraint of storing and performing inference with many distinct models.

In light of the increase in computational cost, it is of great value to understand exactly when we might expect ensembling to improve performance non-trivially. In particular, consider the following practical scenario: a practitioner has trained a single (perhaps large and expensive) model, and would like to know whether they can expect significant gains in performance from training additional models and ensembling them. This question lacks a sufficient answer, both from the theoretical and empirical perspectives, and hence motivates the main question of the present work:

> When are ensembles ***really*** effective?

The present work addresses this question, both theoretically and empirically, under very general conditions. We focus our study on the most popular ensemble classifier—the *majority vote classifier* (Definition 1), which we denote by $h_{\mathrm{MV}}$—although our framework also covers variants such weighted majority vote methods.

**Theoretical results.** Our main theoretical contributions, contained in Section 3, are as follows. First, we formally define the *ensemble improvement rate* (EIR, Definition 3), which measures the decrease in error rate from ensembling, on a relative scale. We then introduce a new condition called *competence* (Assumption 1) that rules out pathological cases, allowing us to prove stronger bounds on the ensemble improvement rate. Specifically, 1) we prove (in Theorem 1) that competent ensembles can never hurt performance, and 2) we prove (in Theorem 2) that the EIR can be upper and lower bounded by linear functions of the *disagreement-error ratio* (DER, Definition 4). ***Our theoretical results predict that ensemble improvement will be high whenever disagreement is large relative to the average error rate (i.e.,*** $\mathrm{DER} > 1$***).*** Moreover, we show (in Appendix A.3) that as Corollaries of our theoretical results, we obtain new bounds on the error rate of the majority vote classifier that significantly improve on previous results, provided the competence assumption is satisfied.

**Empirical results.** In light of our new theoretical understanding of ensembling, we perform a detailed empirical analysis of ensembling in practice. In Section 4, we evaluate the assumptions and predictions made by the theory presented in Section 3. In particular, we verify on a variety of tasks that the competence condition holds, we verify empirically the linear relationship between the EIR and the DER, as predicted by our bounds. In Section 5, we provide significant evidence for distinct behavior arising for ensembles in and out of the "interpolating regime," i.e., when each of the constituent classifiers in an ensemble has sufficient capacity to achieve zero training error. ***We show 1) that interpolating ensembles exhibit consistently lower ensemble improvement rates, and 2) that this corresponds to ensembles transitioning (sometimes sharply) from the regime*** $\mathrm{DER} > 1$ ***to*** $\mathrm{DER} < 1$***.*** Finally, we also show that tree-based ensembles represent a unique exception to this phenomenon, making them particularly well-suited to ensembling.

In addition to the results presented in the main text, we provide supplemental theoretical results (including all proofs) in Appendix A, as well as supplemental empirical results in Appendix B.

## 2 Background and preliminaries

In this section, we present some background as well as preliminary results of independent interest that set the context for our main theoretical and empirical results.

### 2.1 Setup

In this work, we focus on the $K$-class classification setting, wherein the data $(X, Y) \in \mathcal{X} \times \mathcal{Y} \sim \mathcal{D}$ consist of features $\boldsymbol{x} \in \mathcal{X}$ and labels $y \in \mathcal{Y} = \{1, \ldots, K\}$. Classifiers are then functions $h : \mathcal{X} \to \mathcal{Y}$ that belong to some set $\mathcal{H}$. To measure the performance of a single classifier $h$ on the data distribution $\mathcal{D}$, we use the usual error rate:

$$L_{\mathcal{D}}(h) = \mathbb{E}_{X,Y \sim \mathcal{D}}[\mathbb{1}(h(X) \neq Y)].$$

For notational convenience, we will drop the explicit dependence on $\mathcal{D}$ whenever it is apparent from context.

A central object in our study is a distribution $\rho$ over classifiers. Depending on the context, this distribution could represent a variety of different things. For example, $\rho$ could be:

  i) A discrete distribution on a finite set of classifiers $\{h_1, \ldots, h_M\}$ with weights $\rho_1, \ldots, \rho_M$, e.g., representing normalized weights in a weighted ensembling scheme;

 ii) A distribution over parameters $\theta$ of a parametric family of models, $h_\theta$, determined, e.g., by a stochastic optimization algorithm with random initialization;

iii) A Bayesian posterior distribution.

The distribution $\rho$ induces two error rates of interest. The first is the ***average error rate of any single classifier*** under $\rho$, defined to be $\mathbb{E}_{h \sim \rho}[L(h)]$. The second is the ***error rate of the majority vote classifier***, $h_{\mathrm{MV}}$, which is defined for a distribution $\rho$ as follows.

**Definition 1** (Majority vote classifier). Given $\rho$, the ***majority vote classifier*** is the classifier which, for an input $\boldsymbol{x}$, predicts the most probable class for this input among classifiers drawn from $\rho$,

$$h_{\mathrm{MV}}(\boldsymbol{x}) = \arg\max_j \; \mathbb{E}_{h \sim \rho}[\mathbb{1}(h(\boldsymbol{x}) = j)].$$

In the Bayesian context, $\rho = \rho(h \mid X_{\mathrm{train}}, y_{\mathrm{train}})$ is a posterior distribution over classifiers. In this case, the majority vote classifier is often called the Bayes classifier, and the error rate $L(h_{\mathrm{MV}})$ is called the Bayes error rate. In such contexts, the average error rate is often referred to as the Gibbs error rate associated with $\rho$ and $\mathcal{D}$.

Finally, we will present results in terms of the *disagreement rate* between classifiers drawn from a distribution $\rho$, defined as follows.

**Definition 2** (Disagreement). The ***disagreement rate*** between two classifiers $h, h'$ is given by $D_{\mathcal{D}}(h, h') = \mathbb{E}_{X \sim \mathcal{D}}[\mathbb{1}(h(X) \neq h'(X))]$. The ***expected disagreement rate*** is $\mathbb{E}_{h,h' \sim \rho}[D_{\mathcal{D}}(h, h')]$, where $h, h' \sim \rho$ are drawn independently.

**Remark 1.** We focus on the case of classification with the 0/1 loss for a few reasons. First, this scenario represents a significant portion of the practical uses of ensembling. Moreover, it also presents the greatest challenge from a technical perspective, when compared to convex loss functions such as cross entropy, in which case Jensen's inequality immediately implies improvement from ensembling, or regression with squared error, in which case precise analytical results are more easily obtained (see Remark 3 for an example).

### 2.2 Prior work

**Ensembling theory.** Perhaps the simplest general relation between the majority vote error rate and the average error rate guarantees only that the majority vote classifier is no worse than twice the average error rate [LMRR17, MLIS20]. To see this, let $W_\rho \equiv W_\rho(X, Y) = \mathbb{E}_{h \sim \rho}[\mathbb{1}(h(X) \neq Y)]$ denote the proportion of erroneous classifiers in the ensemble for a randomly sampled input-output pair $(X, Y) \sim \mathcal{D}$, and note that $\mathbb{E}[W_\rho] = \mathbb{E}[L(h)]$. Then, by a "first-order" application of Markov's inequality, we have that

$$0 \leq L(h_{\mathrm{MV}}) \leq \mathbb{P}(W_\rho \geq 1/2) \leq 2\,\mathbb{E}[W_\rho] = 2\,\mathbb{E}_{h \sim \rho}[L(h)]. \tag{1}$$

This bound is almost always uninformative in practice. Indeed, it may seem surprising that an ensemble classifier could perform *worse* than the average of its constituent classifiers, much less a factor of two worse. Nonetheless, the first-order upper bound is, in fact, tight: there exist distributions $\rho$ (over classifiers) and $\mathcal{D}$ (over data) such that the majority vote classifier is twice as erroneous as any one classifier, on average. As one might expect, however, this tends to happen only in pathological cases; we give examples of such ensembles in Appendix C.

To circumvent the shortcomings of the simple first-order bound, more recent approaches have developed bounds incorporating "second-order" information from the distribution $\rho$ [MLIS20]. One successful example of this is given by a class of results known as C-bounds [GLL+15, LMRR17]. The most general form of these bounds states, provided $\mathbb{E}[M_\rho(X, Y)] > 0$, that $L(h_{\mathrm{MV}}) \leq 1 - \mathbb{E}[M_\rho(X, Y)]^2 / \mathbb{E}[M_\rho^2(X, Y)]$, where $M_\rho(X, Y) = \mathbb{E}_{h \sim \rho}[\mathbb{1}(h(X) = Y)] - \max_{j \neq Y} \mathbb{E}_{h \sim \rho}[\mathbb{1}(h(X) = j)]$ is called the *margin*. In the binary classification case, the condition $\mathbb{E}[M_\rho(X, Y)] > 0$ is equivalent to the assumption $\mathbb{E}_{h \sim \rho}[L(h)] < 1/2$. Hence, it can be viewed as a requirement that individual classifiers are "weak learners." The same condition is used to derive a very similar bound for random forests in [Bre01], which is then further upper bounded to obtain a more intuitive (though weaker) bound in terms of the "c/s2" ratio. Relatedly, [MLIS20] obtains a bound on the error rate of the majority-vote classifier, in the special case of binary classification, directly in terms of the disagreement rate, taking the form $4\mathbb{E}[L(h)] - 2\mathbb{E}[D(h, h')]$. We note that our theory improves this bound by factor of 2 (see Appendix A.3). Other results obtain similar expressions, but in terms of different loss functions, e.g., cross-entropy loss [ABP+22, OCnM22].

**Other related studies.** In addition to theoretical results, there have been a number of recent empirical studies investigating the use of ensembling. Perhaps the most closely related is [ABPC22], which shows, perhaps surprisingly, that ensembles do not benefit significantly from encouraging diversity during training. In contrast to the present work, [ABPC22] focuses on the cross entropy loss for classification (which facilitates somewhat simpler theoretical analysis), whereas we focus on the more intuitive and commonly used classification error rate. Moreover, while [ABPC22] study the ensemble improvement *gap* (i.e., the difference between the average loss of a single classifier and the ensemble loss), we focus on the gap in error rates on a relative scale. As we show, this provides *much* finer insights into ensembles improvement. To complement this, [FHL19] study ensembling from a loss landscape perspective, evaluating how different approaches to ensembling, such as deep ensembles, Bayesian ensembles, and local methods like Gaussian subspace sampling compare in function and weight space diversity. Other recent work has studied the use of ensembling to provide uncertainty estimates for prediction [LPB17], and to improve robustness to out-of-distribution data [OFR+19], although the ubiquity of these findings has recently been questioned in [ABP+22].

## 3   Ensemble improvement, competence, and disagreement-error ratio

In this section, our goal is to characterize theoretically the rate at which ensembling improves performance. To do this, we first need to formalize a metric to quantify the benefit from ensembling. One natural way of measuring this improvement would be to compute the gap $\mathbb{E}_{h \sim \rho}[L(h)] - L(h_{\mathrm{MV}})$. A similar gap was the focus of [ABPC22], although in terms of the cross-entropy loss, rather than the classification error rate. However, the unnormalized gap can be misleading—in particular, it will tend to be small whenever the average error rate itself is small, thus making it impractical to compare, e.g., across tasks of varying difficulty. Instead, we work with a normalized version of the average-minus-ensemble error rate gap, where the effect of the normalization is to measure this error in a relative scale. We call this the ensemble improvement rate.

**Definition 3** (Ensemble improvement rate). Given distributions $\rho$ over classifiers and $\mathcal{D}$ over data, provided that $\mathbb{E}_{h \sim \rho}[L(h)] \neq 0$, the ***ensemble improvement rate (EIR)*** is defined as

$$\mathrm{EIR} = \frac{\mathbb{E}_{h \sim \rho}[L(h)] - L(h_{\mathrm{MV}})}{\mathbb{E}_{h \sim \rho}[L(h)]}.$$

In contrast to the unnormalized gap, the ensemble improvement rate can be large even for very easy tasks with a small average error rate. Recall the simple first-order bound on the majority-vote classifier: $L(h_{\mathrm{MV}}) \leq 2\mathbb{E}[L(h)]$. Rearranging, we deduce that $\mathrm{EIR} \geq -1$. Unfortunately, in the absence of additional information, this first-order bound is in fact tight: one can construct ensembles for this $L(h_{\mathrm{MV}}) = 2\mathbb{E}[L(h)]$ (see Appendix C). However, this bound is inconsistent with how ensembles generally behave and practice, and indeed it tells us nothing about when ensembling can

improve performance. In the subsequent sections, we derive improved bounds on the EIR that can provide such insight.

## 3.1 Competent ensembles never hurt

Surprisingly, to our knowledge, there is no known characterization of the majority-vote error rate that *guarantees* it can be no worse than the error rate of any individual classifier, on average. Indeed, it turns out this is the result of strange behavior that can arise for particularly pathological ensembles rarely encountered in practice (see Appendix C for a more detailed discussion of this). To eliminate these cases, we introduce a mild condition that we call *competence*.

**Assumption 1** (Competence). Let $W_\rho \equiv W_\rho(X, Y) = \mathbb{E}_{h\sim\rho}[\mathbb{1}(h(X) \neq Y)]$. The ensemble $\rho$ is ***competent*** if for every $0 \leq t \leq 1/2$,

$$\mathbb{P}(W_\rho \in [t, 1/2)) \geq \mathbb{P}(W_\rho \in [1/2, 1-t]).$$

The competence assumption guarantees that the ensemble is not pathologically bad, and in particular it eliminates the scenarios under which the naive first-order bound (1) is tight. As we will show in Section 4, the competence condition is quite mild, and it holds broadly in practice. Our first result uses competence to improve non-trivially the naive first-order bound.

**Theorem 1.** *Competent ensembles never hurt performance, i.e.,* EIR $\geq 0$.

Translated into a bound on the majority vote classifier, Theorem 1 guarantees that $L(h_{\mathrm{MV}}) \leq \mathbb{E}[L(h)]$, improving on the naive first-order bound (1) by a factor of two. To the best of our knowledge, the competence condition is the first of its kind, in that it guarantees what is widely observed in practice, i.e., that ensembling cannot *hurt* performance. However, it is insufficient to answer the question of *how much* ensembling can improve performance. To address this question, we turn to a "second-order" analysis involving the disagreement rate.

## 3.2 Quantifying ensemble improvement with the disagreement-error ratio

Our central result in this section will be to relate the EIR to the ratio of the disagreement to average error rate, which we define formally below.

**Definition 4** (Disagreement-error ratio). Given the distributions $\rho$ over classifiers and $\mathcal{D}$ over data, provided that $\mathbb{E}_{h\sim\rho}[L(h)] \neq 0$, the ***disagreement-error ratio (DER)*** is defined as

$$\mathrm{DER} = \frac{\mathbb{E}_{h,h'\sim\rho}[D(h, h')]}{\mathbb{E}_{h\sim\rho}[L(h)]}.$$

Our next result relates the EIR to a linear function of the DER.

**Theorem 2.** *For any competent ensemble $\rho$ of $K$-class classifiers, provided $\mathbb{E}_{h\sim\rho}[L(h)] \neq 0$, the ensemble improvement rate satisfies*

$$\mathrm{DER} \geq \mathrm{EIR} \geq \frac{2(K-1)}{K}\mathrm{DER} - \frac{3K-4}{K}.$$

Note that neither Theorem 1 nor Theorem 2 is uniformly stronger. In particular, if DER $<$ $(3K - 4)/(2K - 2)$ then the lower bound provided in Theorem 1 will be superior to the one in Theorem 2.

Theorem 2 predicts that the EIR is fundamentally governed by a linear relationship with the DER — a result that we will verify empirically in Section 4. Importantly, we note that there are two distinct regimes in which the bounds in Theorem 2 provide non-trivial guarantees.

> **DER small ($< 1$).** In this case, by the trivial bound (1), EIR $\leq 1$, and thus the upper bound in Theorem 2 guarantees ensemble improvement cannot be too large whenever DER $< 1$, that is, whenever disagreement is small relative to the average error rate.

> **DER large ($> 1$).** In this case, the lower bound in Theorem 2 guarantees ensemble improvement whenever disagreement is sufficiently large relative to the average error rate, in particular when DER $\geq (3K - 4)/(2K - 2) \geq 1$.

In our empirical evaluations, we will see that these two regimes (DER > 1 and DER < 1) strongly distinguish between situations in which ensemble improvement is high, and when the benefits of ensembling are significantly less pronounced.

Moreover, Theorem 2 captures an important subtlety in the relationship between ensemble improvement and predictive diversity. In particular, while general intuition—and a significant body of prior literature, as discussed in Section 2—suggests that higher disagreement leads to high ensemble improvement, this may *not* be the case if the disagreement is nominally large, but small relative to the average error rate.

**Remark 2** (Corollaries of Theorems 1 and 2)**.** Using some basic algebra, the upper and lower bounds presented in Theorems 1 and 2 can easily be translated into upper and lower bounds on the error rate of the majority vote classifier itself. For the sake of space, we defer discussion of these Corollaries to Appendix A.3, although we note that the resulting bounds constitute significant improvements on existing bounds, which we verify both analytically (when possible) and empirically.

**Remark 3** (Relation to the regression case)**.** We remark that the bounds in Theorem 2 share a direct (and simpler) analogue in the regression case with the squared error loss. In this case, a bias-variance decomposition (taken over randomness arising from the sampling of a regressor $f \sim \rho$) yields the identity:

$$\mathbb{E}_{f\sim\rho}[L(f)] = L(\bar{f}) + \mathbb{E}_{X\sim\mathcal{D}}[\mathbb{V}_{f\sim\rho}[f(X)]]$$

where here the ensemble regressor is $\bar{f}(\boldsymbol{x}) = \mathbb{E}_{f\sim\rho}[f(\boldsymbol{x})]$ and $\mathbb{V}[\cdot]$ is the variance. Normalizing, we obtain the identity

$$\mathrm{EIR} = \frac{\mathbb{E}_{f\sim\rho}[L(f)] - L(\bar{f})}{\mathbb{E}_{f\sim\rho}[L(f)]} = \frac{\mathbb{E}_{X\sim\mathcal{D}}[\mathbb{V}_{f\sim\rho}[f(X)]]}{\mathbb{E}_{f\sim\rho}[L(f)]}$$

where the latter variance-average error ratio is a natural analogue to the DER. Note also since the variance is non-negative, this expression immediately implies that $\mathrm{EIR} \geq 0$ in the regression case. See Appendix A.4 for a derivation.

# 4 Evaluating the theory

In this section, we investigate the assumptions and predictions of the theory proposed in Section 3. In particular we will show 1) that the competence assumption holds broadly in practice, across a range of architectures, ensembling methods and datasets, and 2) that the EIR exhibits a close linear relationship with the DER, as predicted by Theorem 2.

Before presenting our findings, we first briefly describe the experimental settings analyzed in the remainder of the paper. Our goal is to select a sufficiently broad range of tasks and methods so as to demonstrate the generality of our conclusions.

## 4.1 Setup for empirical evaluations

In Table 1 we provide a brief description of our experimental setup; more extensive experimental details can be found in Appendix B.1.

Table 1: Datasets and ensembles used in empirical evaluations, where $C$ denotes the number of classes, and $M$ denotes the number of classifiers.

| Datasets | | | Ensembles | | | |
|---|---|---|---|---|---|---|
| **Dataset** | **C** | **Reference** | **Base classifier** | **Ensembling** | **M** | **Reference** |
| MNIST (5K subset) | 10 | [Den12] | ResNet20-Swish | Bayesian Ens. | 100 | [IVHW21] |
| CIFAR-10 | 10 | [KH+09] | ResNet18 | Deep Ens. | 5 | [KH+09] |
| IMDB | 2 | [MDP+11] | CNN-LSTM | Bayesian Ens. | 100 | [IVHW21] |
| QSAR | 2 | [BGCT19] | BERT (fine-tune) | Deep Ens. | 25 | [DCLT19, SYT+22] |
| Thyroid | 2 | [QCHL87] | Random Features | Bagging | 30 | N/A |
| GLUE (7 tasks) | 2-3 | [WSM+19] | Decision Trees | Random Forests | 100 | [PVG+11, Bre01] |

## 4.2 Verifying competence in practice.

Our theoretical results in Section 3 relied on the competence condition. One might wonder whether competent ensembles exist, and if so how ubiquitous they are. Here, we test that assumption. (That is, we test not just the predictions of our theory, but also the assumptions of our theory.)

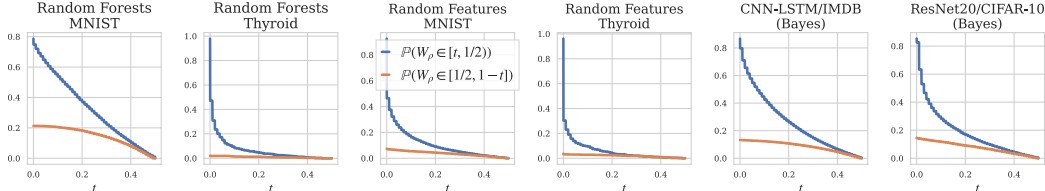

Figure 1: **Verifying the competence assumption in practice.** $W_\rho(X, Y)$ in Assumption 1 is estimated using hold-out data. Across all tasks, ■ > ■, supporting Assumption 1.

We have observed that the competence assumption is empirically very mild, and that in practice it applies very broadly. In Figure 1, we estimate both $\mathbb{P}(W_\rho \in [t, 1/2))$ and $\mathbb{P}(W_\rho \in [1/2, 1-t])$ on test data, validating that competence holds for various types of ensembles across a subset of tasks. To do this, given a test set of examples $\{(\boldsymbol{x}_j, y_j)\}_{j=1}^m$ and classifiers $h_1, \ldots, h_N$ drawn from $\rho$, we construct the estimator

$$\widehat{W}_\rho^{(j)} = \frac{1}{N} \sum_{n=1}^N \mathbb{1}(h_n(\boldsymbol{x}_j) \neq y_j),$$

and we calculate $\mathbb{P}(W_\rho \in [t, 1/2))$ and $\mathbb{P}(W_\rho \in [1/2, 1-t])$ from the empirical CDF of $\{\widehat{W}_\rho^{(j)}\}_{j=1}^m$. In Appendix B, we provide additional examples of competence plots across more experimental settings (and we observe substantially the same results).

### 4.3 The linear relationship between DER and EIR.

Theorem 2 predicts a linear relationship between the EIR and the DER; here we verify that this relationship holds empirically. In Figure 2, we plot the EIR against the DER across several experimental settings, varying capacity hyper-parameters (width for the ResNet18 models, number of random features for the random feature classifiers, and number of leaf nodes for the random forests), reporting the equation of the line of best fit, as well as the Pearson correlation between the two metrics. Across 6 of the 8 experimental settings evaluated, we find a very strong linear relationship – with the Pearson $R \geq 0.96$. The two exceptions are found for the Thyroid classification dataset (though there is still a strong trend between the two quantities, with $R \approx 0.8$).

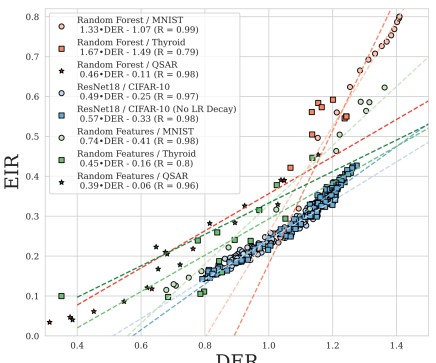

Figure 2: EIR **is linearly correlated with the** DER. We plot the EIR against the DER across a variety of experimental settings, and we observe a close linear relationship between the EIR and DER, as predicted by our theoretical results. In the legend, we also report the equation for the line of best fit within each setting, as well as the Pearson correlation.

Interestingly, we can compare the lines of best fit to the theoretical linear relationship predicted by Theorem 2. In the case of the binary classification datasets (QSAR and Thyroid), the bound predicts that $\text{EIR} \approx \text{DER} - 1$; while for the 10-class problems (MNIST and CIFAR-10), the equation is $\text{EIR} \approx 1.8\,\text{DER} - 2.6$. While for some examples (e.g., random forests on MNIST), the equations are close to those theoretically predicted, there is a clear gap between theory and the experimentally measured relationships for other tasks. In particular, we notice a significant difference in the governing equations for the same datasets between random forests and the random feature ensembles, suggesting that the relationship is to some degree modulated by the model architecture – something our theory cannot capture. We therefore see refining our theory to incorporate such information as an important direction for future work.

## 5 Ensemble improvement is low in the interpolating regime

In this section, we will show that the DER behaves qualitatively differently for interpolating versus non-interpolating ensembles, in particular exhibiting behavior associated with phase transitions. Such phase transitions are well-known in the statistical mechanics approach to learning [EdB01, MM17, TKM21, YHT+21], but they have been viewed as surprising from the more traditional approach to statistical learning theory [BHMM19]. We will use this to understand when ensembling is and is not effective for deep ensembles, and to explain why tree-based methods seem to benefit so much

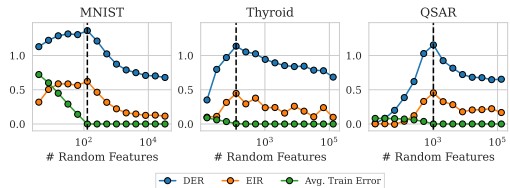

Figure 3: **Bagged random feature classifiers.** Blacked dashed line represents the interpolation threshold. Across all tasks, DER and EIR are maximized at this point, and then decrease thereafter.

Figure 4: **Random forest classifiers.** Blacked dashed line represents the interpolation threshold. Across all tasks, DER and EIR are maximized at this point, and then remain constant thereafter.

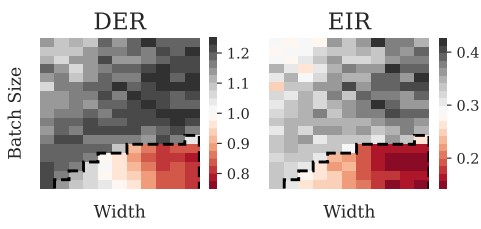

(a) **Without LR decay.**

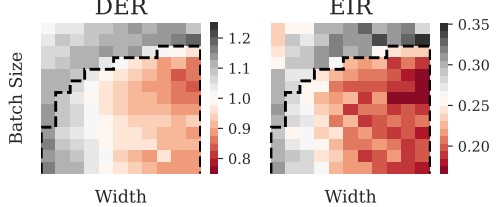

(b) **With LR decay.**

Figure 5: **Large scale studies of deep ensembles on ResNet18/CIFAR-10.** We plot the DER and EIR across a range of hyper-parameters, for two training settings: one with learning rate decay, and one without. The black dashed line indicates the *interpolation threshold*, i.e., the curve below which individual models achieve exactly zero training error. Observe that interpolating ensembles attain distinctly lower EIR than non-interpolating ensembles, and correspondingly have low DER (< 1), compared to non-interpolating ensembles with high DER (> 1).

from ensembling across all settings. We say that a model is *interpolating* if it achieves exactly zero training error, and we say that it is *non-interpolating* otherwise; we call an ensemble interpolating if each of its constituent classifiers is interpolating. Note that for methods that involve resampling of the training data (e.g., bagging methods), we define the training error as the "in-bag" training, i.e., the error evaluated only on the points a classifier was trained on.

**Interpolating random feature classifiers.** We first look at the bagged random feature ensembles on the MNIST, Thyroid, and QSAR datasets. In Figure 3, we plot the EIR, DER and training error for each of these ensembles. We observe the same phenomenon across these three tasks: *as a function of model capacity, the EIR and DER are both maximized at the interpolation threshold, before decreasing thereafter. This indicates that much higher-capacity models, those with the ability to easily interpolate the training data, benefit significantly less from ensembling. In particular, observe that for sufficiently high-capacity ensembles, the DER become less than 1, entering the regime in which our theory guarantees low ensemble improvement.*

**Interpolating deep ensembles.** We next consider the DER and EIR for large-scale empirical evaluations on ResNet18/CIFAR-10 models in batch size/width space, both with and without learning rate decay. See Figure 5 for the results. Note that the use of learning rate decay facilitates easier interpolation of the training data during training, hence broadening the range of hyper-parameters for which interpolation occurs. The figures are colored so that ensembles in the regime DER < 1 are in red, while ensembles with DER > 1 are in grey. The black dashed line indicates the interpolation threshold, i.e., the curve in hyper-parameter space below which ensembles achieve zero training error (meaning every classifier in the ensemble has zero training error). *Observe in particular that, across all settings,* all *models in the regime* DER < 1 *are interpolating models; and, more generally, that interpolating models tend to exhibit* much *smaller* DER *than non-interpolating models.* Observe moreover that there can be a sharp transition between these two regimes, wherein the DER is large just at the interpolation threshold, and then it quickly decreases beyond that threshold. Correspondingly, ensemble improvement is *much* less pronounced for interpolating ensembles versus more traditional non-interpolating ensembles. This is consistent with results previously observed in the literature, e.g., [GJS⁺20]. We remark also that the behavior observed in Figure 5 exhibits the same phases identified



Figure 6: **Bayesian ensembles on IMDB and CIFAR-10.** These provide examples of ensembles which do *not* interpolate the training data, and which have high DER (> 1) and correspondingly high EIR.

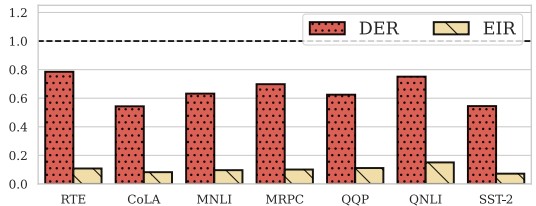

Figure 7: **Ensembles of fine-tuned BERT on GLUE tasks.** Here the models are large relative to the dataset size, and consequently they exhibit low DER (< 1) and EIR across all tasks.

in [YHT+21] (an example of phase transitions in learning more generally [EdB01, MM17]), although the DER itself was not considered in that previous study.

**Bayesian neural networks.**   Next, we show that Bayesian neural networks benefit significantly from ensembling. In Figure 6, we plot the DER and EIR for Bayesian ensembles on the CIFAR-10 and IMDB tasks, using the ResNet20 and CNN-LSTM architectures, respectively. The samples we present here are provided by [INLW21], who use Hamiltonian Monte Carlo to sample accurately from the posterior distribution over models. For both tasks, we observe that both ensembles exhibit DER > 1 and high EIR. In light of our findings regarding ensemble improvement and interpolation, we note that the Bayesian ensembles by design do *not* interpolate the training data (when drawn at non-zero temperature), as the samples are drawn from a distribution not concentrated only on the modes of the training loss. While we do not perform additional experiments with Bayesian neural networks in the present work, evaluating the DER/EIR as a function of posterior temperature for these models is an interesting direction for future work. We hypothesize that the qualitative effective of decreasing the sampling temperature will be similar to that of increasing the batch size in the plots in Figure 5.

**Fine-tuned BERT ensembles.**   Here, we present results for ensembles of BERT models fine-tuned on the GLUE classification tasks. These provide examples of a very large model trained on small datasets, on which interpolation is easily possible. For these experiments, we use 25 BERT models pre-trained from independent initializations provided in [SYT+22]. Each of these 25 models is then fine-tuned on the 7 classification tasks in the GLUE [WSM+19] benchmark set and evaluated on relatively small test sets, ranging in size from  250 (RTE) to  40,000 (QQP) samples. In Figure 7, we plot the EIR and DER across these benchmark tasks and observe that, as predicted, the ensemble improvement rate is low and DER is uniformly low (< 1).

**The unique case of random forests.**   Random forests are one of the most widely-used ensembling methods in practice. Here, we show that the effectiveness of ensembling is much greater for random forest models than for highly-parameterized models like the random feature classifiers and the deep ensembles. Note that for random forests, interpolation of the training data *is* possible, in particular whenever the number of terminal leaf nodes is sufficiently large (where here we again compute the average training error using on the in-bag training examples for each tree), but it is not possible to go "into" the interpolating regime. In Figure 4, we plot the DER, EIR, and training error as a function of the max number of leaf nodes (a measure of model complexity). Before the interpolation threshold, both the EIR and DER increase as a function of model capacity, in line with what is observed for the random feature and deep ensembles. However, we observe distinct behavior at the interpolation threshold: both EIR and DER become constant past this threshold. This is fundamental to tree-based methods, due to the method by which they are fit, e.g., using a standard procedure like CART [BFSO84]. As soon as a tree achieves zero training error, any impurity method used to split the nodes further is saturated at zero, and therefore the models cannot continue to grow. This indicates that trees are particularly well-suited to ensembling across all hyper-parameter values, in contrast to other parameterized types of classifiers.

## 6   Discussion and conclusion

To help answer the question of when ensembling is effective, we introduce the ensemble improvement rate (EIR), which we then study both theoretically and empirically. Theoretically, we provide a

comprehensive characterization of the EIR in terms of the disagreement-error ratio DER. The results are based on a new, mild condition called *competence*, which we introduce to rule out pathological cases that have hampered previous theoretical results. Using a simple first-order analysis, we show that the competence condition is sufficient to guarantee that ensembling cannot hurt performance—something widely observed in practice, but surprisingly unexplained by existing theory. Using a second-order analysis, we are able to theoretically characterize the EIR, by upper and lower bounding it in terms of a linear function of the DER. On the empirical side, we first verify the assumptions of our theory (namely that the competence assumption holds broadly in practice), and we show that our bounds are indeed descriptive of ensemble improvement in practice. We then demonstrate that improvement decreases precipitously for interpolating ensembles, relative to non-interpolating ones, providing a very practical guideline for when to use ensembling.

Our work leaves many directions to explore, of which we name a few promising ones. First, while our theory represents a significant improvement on previous results, there are still directions to extend our analysis. For example, Figure 2 suggests that the relationship between EIR and DER can be even more finely characterized. Is it possible to refine our analysis further to incorporate information about the data and/or model architecture? Second, can we formalize the connection between ensemble effectiveness and the interpolation point, and relate it to similar ideas in the literature?

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
