# A Proofs of our main results

In this section, we provide proofs for our main results. Throughout the section, we denote $\mathbb{P}_{h,h'\sim\rho^2}$, $\mathbb{P}_{h\sim\rho}$, $\mathbb{E}_{h\sim\rho}$ by $\mathbb{P}_{h,h'}$, $\mathbb{P}_h$, $\mathbb{E}_h$, respectively. We also denote $W_\rho(X,Y)$ simply by $W_\rho$. We typically omit explicit dependence on the data distribution $\mathcal{D}$ when it is apparent from context.

## A.1 Proof of Theorem 1

We first state and prove two lemmas that will be used in the proof of Theorem 1. Our first lemma states that majority-vote error $L_\mathcal{D}[h_{\mathrm{MV}}]$ is upper bounded by probability of $W_\rho(X,Y)$ being large.

**Lemma 1.** *There is the inequality* $L_\mathcal{D}[h_{\mathrm{MV}}] \leq \mathbb{P}_\mathcal{D}(W_\rho(X,Y) \geq 1/2)$, *where* $L_\mathcal{D}(h) = \mathbb{E}_\mathcal{D}[\mathbb{1}(h(X) \neq Y)]$, $h_{\mathrm{MV}}(\boldsymbol{x}) = \arg\max_j \mathbb{E}_h[\mathbb{1}(h(\boldsymbol{x}) = j)]$ *and* $W_\rho(X,Y) = \mathbb{E}_h[\mathbb{1}(h(X) \neq Y)]$.

*Proof.* For given data point $x$, $W_\rho < 1/2$ implies that we are predicting the true label correctly more than half of the time. Thus, the majority vote classifier will correctly predict the label on the data point. $\square$

Our next lemma states a property of competent classifiers which plays a crucial role in the main proof.

**Lemma 2.** *Under Assumption 1 (competence), for any increasing function $h$ satisfying $h(0) = 0$,*

$$\mathbb{E}_\mathcal{D}[h(W_\rho)\mathbb{1}_{W_\rho<1/2}] \geq \mathbb{E}_\mathcal{D}[h(\bar{W}_\rho)\mathbb{1}_{\bar{W}_\rho\leq1/2}],$$

*where* $\bar{W}_\rho = 1 - W_\rho$.

*Proof.* For every $x \in [0,1]$,

$$\mathbb{P}_\mathcal{D}(W_\rho\mathbb{1}_{W_\rho<1/2} \geq x) = \mathbb{P}_\mathcal{D}(W_\rho \in [x,1/2))\,\mathbb{1}_{x\leq1/2},$$
$$\mathbb{P}_\mathcal{D}(\bar{W}_\rho\mathbb{1}_{\bar{W}_\rho\leq1/2} \geq x) = \mathbb{P}_\mathcal{D}(\bar{W}_\rho \in [x,1/2])\,\mathbb{1}_{x\leq1/2} = \mathbb{P}_\mathcal{D}(W_\rho \in [1/2,1-x])\,\mathbb{1}_{x\leq1/2}.$$

From Assumption 1, this implies that $\mathbb{P}_\mathcal{D}(W_\rho\mathbb{1}_{W_\rho<1/2} \geq x) \geq \mathbb{P}_\mathcal{D}(\bar{W}_\rho\mathbb{1}_{\bar{W}_\rho\leq1/2} \geq x)$ for all $x \in [0,1]$. Therefore, for any increasing function $h$ satisfying $h(0) = 0$, since $h(x\,\mathbb{1}_{x\leq c}) = h(x)\mathbb{1}_{x\leq c}$,

$$\mathbb{P}_\mathcal{D}(h(W_\rho)\mathbb{1}_{W_\rho<1/2} \geq x) \geq \mathbb{P}_\mathcal{D}(h(\bar{W}_\rho)\mathbb{1}_{\bar{W}_\rho\leq1/2} \geq x).$$

As $W_\rho$ is non-negative, the equality $\mathbb{E}X = \int_0^\infty \mathbb{P}(X \geq x)\mathrm{d}x$ concludes the proof. $\square$

With these two lemmas, we now provide the proof of Theorem 1.

*Proof of Theorem 1.* From Lemma 1 and the relation $\mathbb{E}_h[L_\mathcal{D}(h)] = \mathbb{E}_\mathcal{D}[W_\rho]$ (Fubini's theorem), it suffices to show that $\mathbb{P}_\mathcal{D}(W_\rho \geq 1/2) \leq \mathbb{E}_\mathcal{D}[W_\rho]$. To do so, observe

$$\mathbb{E}_\mathcal{D}[(W_\rho-1)\mathbb{1}_{W_\rho\geq1/2}] + \mathbb{E}_\mathcal{D}[\bar{W}_\rho\mathbb{1}_{\bar{W}_\rho\leq1/2}] = \mathbb{E}_\mathcal{D}[(W_\rho-1)\mathbb{1}_{W_\rho\geq1/2}] + \mathbb{E}_\mathcal{D}[(1-W_\rho)\mathbb{1}_{W_\rho\geq1/2}] = 0.$$

Applying Lemma 2 with $h(x) = x$,

$$\mathbb{E}_\mathcal{D}[W_\rho] - \mathbb{P}_\mathcal{D}(W_\rho \geq 1/2) \geq \mathbb{E}_\mathcal{D}[(W_\rho-1)\mathbb{1}_{W_\rho\geq1/2}] + \mathbb{E}_\mathcal{D}[W_\rho\mathbb{1}_{W_\rho<1/2}]$$
$$\geq \mathbb{E}_\mathcal{D}[(W_\rho-1)\mathbb{1}_{W_\rho\geq1/2}] + \mathbb{E}_\mathcal{D}[\bar{W}_\rho\mathbb{1}_{\bar{W}_\rho\leq1/2}] = 0.$$

which proves

$$L_\mathcal{D}[h_{\mathrm{MV}}] \underset{\mathrm{Lemma1}}{\leq} \mathbb{P}_\mathcal{D}(W_\rho \geq 1/2) \leq \mathbb{E}_\mathcal{D}[W_\rho] = \mathbb{E}_h[L_\mathcal{D}(h)]. \tag{2}$$

This implies EIR $\geq 0$. $\square$

### A.2 Proof of Theorem 2

#### A.2.1 Lower bound of EIR

To prove the lower bound, we first define the tandem loss, as used in [MLIS20].

**Definition 5** (Tandem loss). *Define the tandem loss to be $L(h, h') = \mathbb{E}_{\mathcal{D}}[\mathbb{1}(h(X) \neq Y)\mathbb{1}(h'(X) \neq Y)]$.*

We also rely on the following lemma, which appears as Lemma 2 in [MLIS20]. It provides the connection between the average error rate for each data point, $W_\rho$ and the tandem loss, $L(h, h')$.

**Lemma 3.** *The equality $\mathbb{E}_{\mathcal{D}}[W_\rho{}^2] = \mathbb{E}_{h,h'}[L(h, h')]$ holds.*

We first state and prove the following lemma, which provides an upper bound on the tandem loss.

**Lemma 4.** *For the $K$-class problem,*

$$\mathbb{E}_{h,h'}[L(h, h')] \leq \frac{2(K-1)}{K}\left(\mathbb{E}_h[L(h)] - \frac{1}{2}\mathbb{E}_{h,h'}[D(h, h')]\right).$$

*Proof.* We denote $\mathbb{P}_h(h(X) \neq Y)$ by $\bar{h}_Y(X)$. Note that $\mathbb{E}_{\mathcal{D}}(1 - \bar{h}_Y(X)) = \mathbb{E}_h[L(h)]$ and

$$\mathbb{E}_{h,h'}[L(h, h')] = \mathbb{E}_{\mathcal{D}}[\mathbb{P}_h(h(X) \neq Y)\mathbb{P}_{h'}(h'(X) \neq Y)]$$
$$= \mathbb{E}_{\mathcal{D}}[(1 - \bar{h}_Y(X))^2].$$

Then we get

$$\mathbb{E}_{h,h'}[L(h, h')] = \mathbb{E}_{\mathcal{D}}[(1 - \bar{h}_Y(X))^2]$$
$$= 1 - \mathbb{E}_{\mathcal{D}}[\bar{h}_Y(X)] - \mathbb{E}_{\mathcal{D}}[\bar{h}_Y(X)(1 - \bar{h}_Y(X))]$$
$$= \mathbb{E}_h[L(h)] - \mathbb{E}_{\mathcal{D}}[\bar{h}_Y(X)(1 - \bar{h}_Y(X))].$$

Now we will derive a lower bound of the second term. Since

$$\mathbb{E}_{h,h'}[\mathbb{1}(h(X) \neq h'(X))] = \sum_j \bar{h}_j(X)(1 - \bar{h}_j(X)),$$

it follows that

$$\bar{h}_Y(X)(1 - \bar{h}_Y(X)) = \mathbb{E}_{h,h'}[\mathbb{1}(h(X) \neq h'(X))] - \sum_{j \neq Y} \bar{h}_j(X)(1 - \bar{h}_j(X)).$$

By maximizing $\sum_{j \neq Y} \bar{h}_j(X)(1 - \bar{h}_j(X))$ subject to $\sum_{j \neq Y} \bar{h}_j(X) = 1 - \bar{h}_Y(X)$, we get $\bar{h}_j(X) = \frac{1 - \bar{h}_Y(X)}{K-1}$, which yields the upper bound

$$\sum_{j \neq Y} \bar{h}_j(X)(1 - \bar{h}_j(X)) \leq \frac{K-2}{K-1}(1 - \bar{h}_Y(X)) + \frac{1}{K-1}\bar{h}_Y(X)(1 - \bar{h}_Y(X)).$$

It follows that

$$\mathbb{E}_{\mathcal{D}}[\bar{h}_Y(X)(1 - \bar{h}_Y(X))] \geq \frac{K-1}{K}\mathbb{E}_{h,h'}[D(h, h')] - \frac{K-2}{K}\mathbb{E}_h[L(h)],$$

and thus that

$$\mathbb{E}_{h,h'}[L(h, h')] = \mathbb{E}_h[L(h)] - \mathbb{E}_{\mathcal{D}}[\bar{h}_Y(X)(1 - \bar{h}_Y(X))]$$
$$\leq \mathbb{E}_h[L(h)] - \left(\frac{K-1}{K}\mathbb{E}_{h,h'}[D(h, h')] - \frac{K-2}{K}\mathbb{E}_h[L(h)]\right)$$
$$= \frac{2(K-1)}{K}\left(\mathbb{E}_h[L(h)] - \frac{1}{2}\mathbb{E}_{h,h'}[D(h, h')]\right).$$

$\square$

We now provide the proof for the lower bound of EIR in Theorem 2.

*Proof.* We first claim that $\mathbb{P}_{\mathcal{D}}(W_\rho \geq 1/2) \leq 2\,\mathbb{E}_{\mathcal{D}}[W_\rho{}^2]$. Then, we have

$$\mathbb{E}_{\mathcal{D}}\big[(2W_\rho{}^2 - 1)\mathbb{1}_{W_\rho \geq 1/2}\big] = \mathbb{E}_{\mathcal{D}}\big[(2(1 - \bar{W}_\rho)^2 - 1)\mathbb{1}_{\bar{W}_\rho \leq 1/2}\big]$$
$$= \mathbb{E}_{\mathcal{D}}\big[(1 - 4\bar{W}_\rho + 2\bar{W}_\rho{}^2)\mathbb{1}_{\bar{W}_\rho \leq 1/2}\big],$$

where $\bar{W}_\rho = 1 - W_\rho$. Therefore,

$$\mathbb{E}_{\mathcal{D}}\big[(2W_\rho{}^2 - 1)\mathbb{1}_{W_\rho \geq 1/2}\big] + \mathbb{E}_{\mathcal{D}}\big[2\bar{W}_\rho{}^2\mathbb{1}_{\bar{W}_\rho \leq 1/2}\big] = \mathbb{E}_{\mathcal{D}}\big[(1 - 4\bar{W}_\rho + 4\bar{W}_\rho{}^2)\mathbb{1}_{\bar{W}_\rho \leq 1/2}\big] \geq 0.$$

Now we apply Lemma 2 with $h(x) = 2x^2$, to obtain

$$\mathbb{E}_{\mathcal{D}}[2W_\rho{}^2] - \mathbb{P}_{\mathcal{D}}(W_\rho \geq 1/2) \geq \mathbb{E}_{\mathcal{D}}\big[(2W_\rho{}^2 - 1)\mathbb{1}_{W_\rho \geq 1/2}\big] + \mathbb{E}_{\mathcal{D}}\big[2W_\rho{}^2\mathbb{1}_{W_\rho < 1/2}\big]$$
$$\geq \mathbb{E}_{\mathcal{D}}\big[(2W_\rho{}^2 - 1)\mathbb{1}_{W_\rho \geq 1/2}\big] + \mathbb{E}_{\mathcal{D}}\big[2\bar{W}_\rho{}^2\mathbb{1}_{\bar{W}_\rho \leq 1/2}\big] \geq 0, \tag{3}$$

which proves the claim, $\mathbb{P}_{\mathcal{D}}(W_\rho \geq 1/2) \leq 2\,\mathbb{E}_{\mathcal{D}}[W_\rho{}^2]$.

Now we put the claim together with Lemmas 1, 3, and 4 to conclude the proof.

$$L(h_{\mathrm{MV}}) \underset{\text{Lemma 1}}{\leq} \mathbb{P}_{\mathcal{D}}(W_\rho \geq 1/2) \leq 2\,\mathbb{E}_{\mathcal{D}}[W_\rho{}^2] \underset{\text{Lemma 3}}{=} 2\,\mathbb{E}_{h,h'}[L(h,h')]$$
$$\underset{\text{Lemma 4}}{\leq} \frac{4(K-1)}{K}\left(\mathbb{E}_h[L(h)] - \frac{1}{2}\mathbb{E}_{h,h'}[D(h,h')]\right) \tag{4}$$

Rearranging the terms, we obtain

$$\mathbb{E}_h[L_{\mathcal{D}}(h)] - L(h_{\mathrm{MV}}) \geq \frac{2(K-1)}{K}\mathbb{E}_{h,h'}[D(h,h')] - \frac{3K-4}{K}\mathbb{E}_h[L(h)]. \tag{5}$$

Dividing the both terms by $\mathbb{E}_h[L(h)]$ gives the lower bound $\frac{2(K-1)}{K}\mathrm{DER} - \frac{3K-4}{K}$. $\qquad\square$

### A.2.2 Upper bound of EIR

We denote $\mathbb{P}_h(h(X) \neq Y)$ by $\bar{h}_Y(X)$. We have

$$\mathbb{E}_h[L(h)] - L(h_{\mathrm{MV}}) = \mathbb{E}_{h,\mathcal{D}}[\mathbb{1}(h(X) \neq Y) - \mathbb{1}(h_{\mathrm{MV}}(X) \neq Y)].$$

Now

$$\mathbb{1}(h(X) \neq Y) - \mathbb{1}(h_{\mathrm{MV}}(X) \neq Y) = \mathbb{1}(h_{\mathrm{MV}}(X) = Y) - \mathbb{1}(h(X) = Y)$$
$$= \mathbb{1}(h(X) \neq h_{\mathrm{MV}}(X))\left(\mathbb{1}(h_{\mathrm{MV}}(X) = Y) - \mathbb{1}(h(X) = Y)\right)$$
$$\leq \mathbb{1}(h(X) \neq h_{\mathrm{MV}}(X)).$$

Now notice $\mathbb{E}_{h,\mathcal{D}}[\mathbb{1}(h(X) \neq h_{\mathrm{MV}}(X))] = 1 - \mathbb{E}_{\mathcal{D}}[\max_k \bar{h}_k(X)]$. Moreover, by Hölder's inequality,

$$\|\bar{\boldsymbol{h}}(X)\|_2^2 \leq \max_k \bar{h}_k(X),$$

and so

$$\mathbb{E}_h[L(h)] - L(h_{\mathrm{MV}}) \leq 1 - \mathbb{E}_{\mathcal{D}}[\max_k \bar{h}_k(X)]$$
$$\leq 1 - \mathbb{E}_{\mathcal{D}}[\|\bar{\boldsymbol{h}}(X)\|^2] = \mathbb{E}_{h,h'}[D(h,h')]. \tag{6}$$

Dividing the both terms by $\mathbb{E}_h[L(h)]$ gives the upper bound DER.

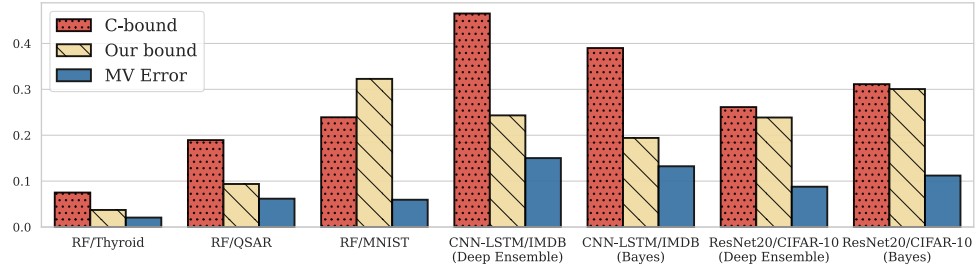

Figure 8: **Our bound** (7) **versus the multi-class C-bound** (9).

### A.3   Upper and lower bounds on the error rate of the majority vote classifier

We now present upper and lower bound on the majority vote classifier that follow from the bounds in Theorem 1 and 2, and compare them with existing bounds in the literature.

**Theorem 3.** *For any competent ensemble $\rho$ of $K$-class classifiers, the majority vote error rate satisfies*

$$L(h_{\mathrm{MV}}) \leq \min\left\{\frac{4(K-1)}{K}\left(\mathbb{E}_{h\sim\rho}[L(h)] - \frac{1}{2}\mathbb{E}_{h,h'\sim\rho}[D(h,h')]\right), \mathbb{E}_{h\sim\rho}[L(h)]\right\}$$
$$L(h_{\mathrm{MV}}) \geq \mathbb{E}_{h\sim\rho}[L(h)] - \mathbb{E}_{h,h'\sim\rho}[D(h,h')].$$

*Proof.* The upper bound follows from inequality (2) and (4). The lower bound follows from inequality (6). □

We have already discussed that the bound $L(h_{\mathrm{MV}}) \leq \mathbb{E}[L(h)]$ represents an improvement by a factor of 2 over the naive first-order bound (1). Here, we further compare the bound

$$L(h_{\mathrm{MV}}) \leq \frac{4(K-1)}{K}\left(\mathbb{E}_{h\sim\rho}[L(h)] - \frac{1}{2}\mathbb{E}_{h,h'\sim\rho}[D(h,h')]\right) \tag{7}$$

to other known results in the literature. The closest in form is a bound specialized to binary case from [MLIS20], which gives

$$L(h_{\mathrm{MV}}) \leq 4\mathbb{E}_{h\sim\rho}[L(h)] - 2\mathbb{E}_{h,h'\sim\rho}[D(h,h')]. \tag{8}$$

Note that plugging in $K = 2$ to (7), we obtain the bound $2\mathbb{E}_{h\sim\rho}[L(h)] - \mathbb{E}_{h,h'\sim\rho}[D(h,h')]$, immediately improving on (8) by a factor of 2 (interestingly, the same factor that we save on the first-order bound). Hence, provided the competence assumption holds, our bound is a direct improvement on this bound, and furthermore generalizes directly to the $K$-class setting.

To our knowledge, the sharpest known upper bound on the majority-vote classifier is the general form of the C-bound given in [LMRR17], which states, provided $\mathbb{E}[M_\rho(X,Y)] > 0$,

$$L(h_{\mathrm{MV}}) \leq 1 - \frac{\mathbb{E}[M_\rho(X,Y)]^2}{\mathbb{E}[M_\rho^2(X,Y)]}, \tag{9}$$

where $M_\rho(X,Y) = \mathbb{E}_{h\sim\rho}[\mathbb{1}(h(X) = Y)] - \max_{j\neq Y}\mathbb{E}_{h\sim\rho}[\mathbb{1}(h(X) = j)]$ is called the *margin*. Unfortunately, the use of the margin function makes direct analytical comparison to our bound difficult. However, the bounds can be compared empirically, where the relevant quantities are estimated on hold-out data. In Figure 8, we compare the value of our bound against the value of the multi-class C-bound, on tasks for which we have verified the competence assumption holds. We find that in all but one case (random forests with MNIST), our bound is superior empirically, sometimes significantly. Interestingly, we observe that our bound does particularly well on tasks with only a few classes. This behavior might be attributed to the constant $\frac{4(K-1)}{K}$ in the upper bound (7) which increases as the number of classes $K$ grows.

## A.4 Exact characterization of the EIR in the regression case

Here we briefly derive an analogous expression for the EIR in the regression case, as mentioned in Remark 3. In this case models are regressors $f$ drawn from a distribution $\rho$, and the natural ensemble model is $\bar{f}(\boldsymbol{x}) = \mathbb{E}_{f\sim\rho}[f(\boldsymbol{x})]$. Here we use as a loss function the squared error $L(f) = \mathbb{E}_{X,Y}[(f(X) - Y)^2]$. We now derive a bias-variance-like expression relating the the average error rate to the ensemble error rate.

**Lemma 5.** *For any distribution $\rho$ over regressors, the average error $\mathbb{E}_{f\sim\rho}[L(f)]$ satisfies*

$$\mathbb{E}_{f\sim\rho}[L(f)] = L(\bar{f}) + \mathbb{E}_X[\mathbb{V}_{f\sim\rho}[f(X)]].$$

*Proof.* The proof is a simple bias-variance decomposition, taken with respect to the randomness in models $f$ drawn from $\rho$. We have

$$\mathbb{E}_f[L(f)] \stackrel{(\star)}{=} \mathbb{E}_{X,Y}[\mathbb{E}_f[(Y - f(X))^2]]$$
$$= \mathbb{E}_{X,Y}\Big[\mathbb{E}_f[(Y - \bar{f}(X) + \bar{f}(X) - f(X))^2]\Big]$$
$$= \mathbb{E}_{X,Y}\Big[\mathbb{E}_f[(Y - \bar{f}(X))^2] + 2\mathbb{E}_f[(Y - \bar{f}(X))(\bar{f}(X) - f(X))] + \mathbb{E}_f[(\bar{f}(X) - f(X))^2]\Big]$$
$$= L(\bar{f}) + \mathbb{E}_X[\mathbb{V}_{f\sim\rho}[f(X)]] + 2\mathbb{E}_f[(Y - \bar{f}(X))(\bar{f}(X) - f(X))],$$

where in $(\star)$ we use Fubini's theorem to exchange integrals. To complete the proof, we claim that the last term is equal to zero. Indeed,

$$\mathbb{E}_f[(Y - \bar{f}(X))(\bar{f}(X) - f(X))] = Y\bar{f}(X) - Y\mathbb{E}_f[f(X)] - \bar{f}^2(X) + \bar{f}(X)\mathbb{E}_f[f(X)]$$
$$= Y\bar{f}(X) + Y\bar{f}(X) - \bar{f}^2(X) + \bar{f}^2(X) = 0.$$

$\square$

Now if we analogously define the EIR in the regression case as

$$\mathrm{EIR} = \frac{\mathbb{E}_f[L(f)] - L(\bar{f})}{\mathbb{E}_f[L(f)]}$$

then we obtain directly the following corollary analogous to Theorem 2.

**Corollary 1.** *The EIR in the regression case satisfies*

$$\mathrm{EIR} = \frac{\mathbb{E}_f[L(f)] - L(\bar{f})}{\mathbb{E}_f[L(f)]} = \frac{\mathbb{E}_X[\mathbb{V}_{f\sim\rho}[f(X)]]}{\mathbb{E}_f[L(f)]}.$$

Note here the variance-average error ratio $\mathbb{E}_X[\mathbb{V}_{f\sim\rho}[f(X)]]/\mathbb{E}_f[L(f)]$ plays the role of the DER in the classification case.

# B Additional empirical results

## B.1 Experimental details

**Bagged random feature classifiers.** We consider ensembles of random ReLU feature classifiers, constructed as follows. For each classifier, we draw a random matrix $\boldsymbol{U} \in \mathbb{R}^{N\times d}$, whose rows $\boldsymbol{u}_j$ are drawn from the uniform distribution on the sphere $\mathbb{S}^{d-1}$. For a given input $\boldsymbol{x} \in \mathbb{R}^d$, we compute the feature $\boldsymbol{z}(\boldsymbol{x}) = \sigma(\boldsymbol{U}\boldsymbol{x})$ where $\sigma(t) = \max(t, 0)$ is the ReLU function. We then fit a multi-class logistic regression model in `scikit-learn` [PVG+11] using these (random) features. To form an ensemble of these classifiers, we additionally perform bagging, by sampling a different set of size $n$ with replacement from the training set of size $n$, independently for each individual classifier. Thus, each classifier is subject to two different types of randomness: the randomness from the sampling of the feature matrix $\boldsymbol{U}$; and the randomness from the bootstrapping of the training data. For the models shown in the competence plot in Figure 1, we use 500 random features and $M = 100$ classifiers.

**Random forests.** We consider random forest (RF) models as implemented in `scikit-learn` [PVG+11], each made up of 20 individual decision trees. We vary the maximum number of leaf nodes in each tree to construct models with varying performance. For the single-ensemble results presented, we use the default parameters implemented in `scikit-learn`. For the random forests, we use a small version of the MNIST dataset with 5000 randomly selected training examples (500 from each of the 10 classes). We also use two binary classification datasets retrieved from the UCI repository [DG17]: the QSAR oral toxicity dataset (7.2k train, 1.8k test examples, 1024 features) [BGCT19]; and the Thyroid disease dataset (2.5k train, 633 test examples, 21 features) [QCHL87]. For the models shown in the competence plot in Figure 1, we use the default settings of the random forest implementation in `scikit-learn`.

**Deep ensembles.** We consider four different architectures for our deep ensembles. We use a standard ResNet18 models [HZRS16] trained on the CIFAR-10 dataset [KH+09], using 100 epochs of SGD with momentum $0.9$, weight decay of $5 \times 10^{-4}$ and a learning rate of $0.1$, while varying the batch size and width hyper-parameters. We report results from two variants of this empirical evaluation: one in which we employ learning rate decay (by dropping the learning rate to $0.01$ after 75 epochs); and another in which we disable learning rate decay. For each setting, we train 5 models from independent initialization to form the respective ensembles. We also evaluate these models on two out-of-distribution databases: CIFAR-10.1 and CIFAR-10-C [RRSS19, HD19] (the latter is itself comprised of 19 different datasets employing various types of data corruption). Finally, we evaluate deep ensembles of 25 standard BERT models [DCLT19], provided with the paper [SYT+22], fine-tuned on the GLUE classification tasks [WSM+19].

**Bayesian ensembles.** For the Bayesian ensembles used in this paper, we consider samples provided in [IVHW21], obtained via large-scale sampling from a Bayesian posterior using Hamiltonian Monte Carlo. To our knowledge, these samples are the most precisely representative of a theoretical Bayesian neural network posterior publicly available. In particular, we use samples on the CIFAR-10 datasets with a ResNet20 architecture, and the IMDB dataset on the CNN-LSTM architecture. We defer to the original paper [IVHW21] for additional details.

### B.2 More competence plots

In this section, we provide additional empirical results.

To further verify that the competence assumption holds broadly in practice, here we include several more examples of competence plots for experiments presented in the main text.

**ResNet18 on CIFAR-10 OOD variants.** In Figures 9 and 10, we plot competence plots for the ResNet18 ensembles on the CIFAR-10, CIFAR-10.1 and a subset of the CIFAR-10-C datasets [RRSS19, HD19]. We find that the competence assumption holds across all examples.

**Fine-tuned BERT models.** In Figure 11, we provide competence plots for the BERT/GLUE fine-tuning tasks. For the RTE, CoLA, MNLI, QQP and QNLI tasks, we find that the competence assumption holds. However, we find two examples here where it does not: the MRPC and SST-2 tasks, although the extent to which the assumption is violated in minor. Since these are particularly small datasets, this may also be a product of noise from low sample size.

## C Pathological ensembles satisfying $L(h_{\mathrm{MV}}) = 2\mathbb{E}[L(h)]$

In this section, we provide two pathological examples of ensembles that makes the "first-order" upper bound tight. In particular, the second example shows that positive margin condition, i.e., $\mathbb{E}[M_\rho(X, Y)] > 0$ where $M_\rho(X, Y) = \mathbb{E}_{h\sim\rho}[\mathbb{1}(h(X) = Y)] - \max_{j\neq Y}\mathbb{E}_{h\sim\rho}[\mathbb{1}(h(X) = j)]$, from existing literature is not enough to rule out pathological cases. Recall that the first-order bound introduced in Section 2.2 is the following:

$$0 \le L(h_{\mathrm{MV}}) \le \mathbb{P}(W_\rho \ge 1/2) \le 2\mathbb{E}[W_\rho] = 2\mathbb{E}_{h\sim\rho}[L(h)].$$

**Example 1** (The first-order upper bound is tight). Consider a classification problem with two classes. For given $\epsilon > 0$, suppose slightly less than half, $0.5 - \epsilon$, fraction of classifiers are the perfect classifier, correctly classifying test data with probability 1, and the other $0.5 + \epsilon$ fraction of classifiers are completely wrong, incorrectly predicting on test data with probability 1. With this composition of classifiers, the average error rate is $0.5 + \epsilon$ and the majority vote error rate is 1. Taking $\epsilon \to 0$ concludes that the first-order upper bound (1) is tight. A visual illustration of the composition of classifiers is given in Figure 12a.

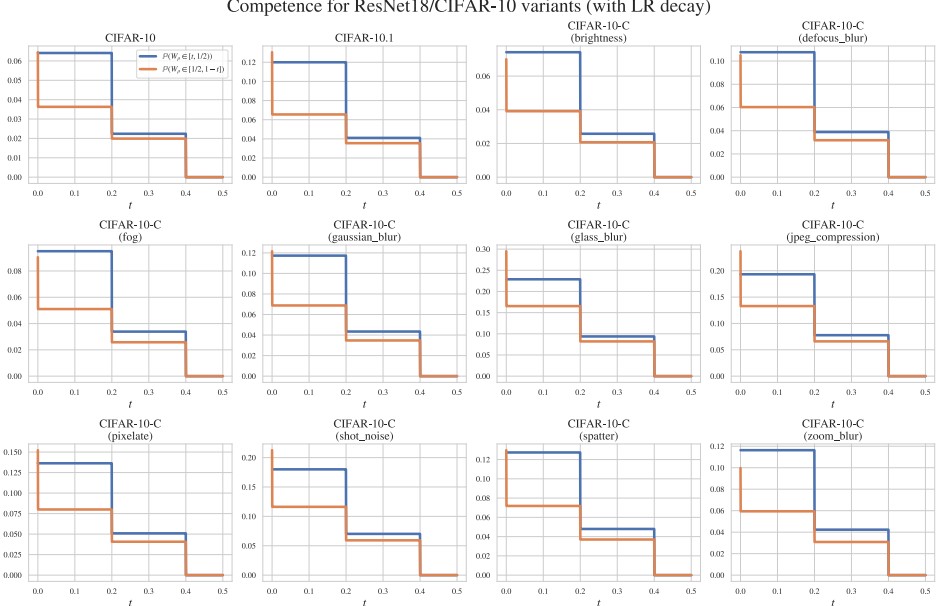

Figure 9: **Competence for ResNet18/CIFAR-10 variants (models with learning rate decay).** We observe that the competence assumption holds across all tasks.

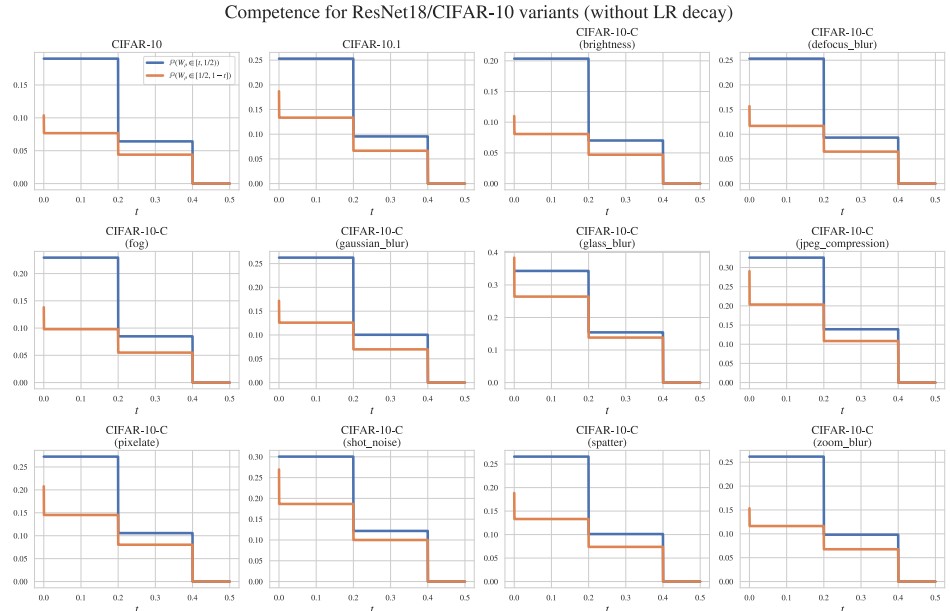

Figure 10: **Competence for ResNet18/CIFAR-10 variants (models without learning rate decay).** We observe that the competence assumption holds across all tasks.

The condition $\mathbb{E}[M_\rho(X,Y)] > 0$ rules out the ensemble described in Example 1. Nonetheless, the first-order bound $2\mathbb{E}[L(h)]$ is tight *even when* $\mathbb{E}[M_\rho(X,Y)] > 0$ is satisfied, as we show with the following example.

**Example 2** (The first-order upper bound is tight even when the margin is large). We again consider a classification problem with two classes. For given $\epsilon > 0$, as in Example 1, slightly less than half, $0.5 - \epsilon$, fraction of classifiers are the perfect classifier. All of the other $0.5 + \epsilon$ fraction of classifiers, on the contrary, now correctly predict on the same $1 - 2\delta$ fraction of the test data and incorrectly predict on the other $2\delta$ fraction of the test data. With this composition of classifiers, the majority vote

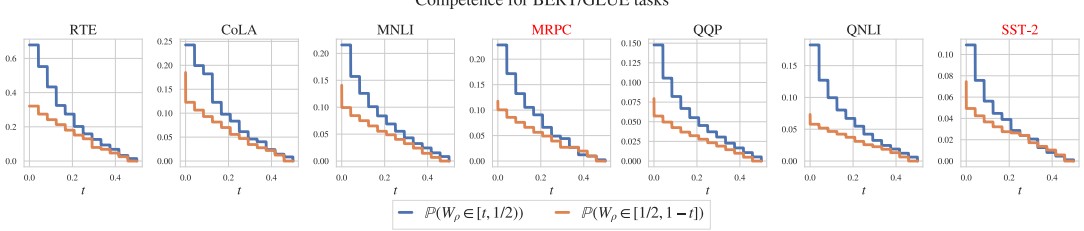

Figure 11: **Competence for BERT/GLUE fine-tuning tasks.** The competence assumption holds for the RTE, CoLA, MNLI, QQP and QNLI tasks, though interestingly, we find that the competence assumption is (to a small degree) violated for two of the tasks: MRPC and SST-2.

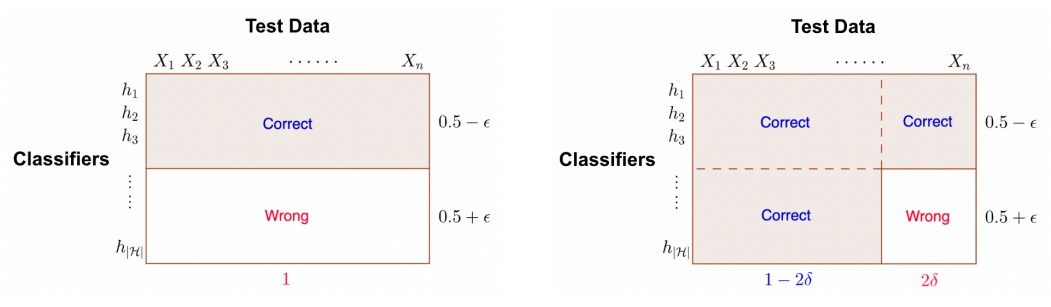

(a) Composition of classifiers in Example 1       (b) Composition of classifiers in Example 2

Figure 12: **Illustration of the composition of classifiers given in Examples 1 and 2.** On each plot, the area of the white box equals to the average test error rate. On Figure 12a, the majority vote error rate is 1, while the average test error rate is $0.5 + \epsilon$. On Figure 12b, the majority vote error rate is $2\delta$ (Rightmost $2\delta$ test data) while the average test error rate is $\delta(1 + 2\epsilon)$. The margin of each composition of classifiers is $2\epsilon \rightarrow 0$ and $1 - 2\delta(1 + 2\epsilon) > 0$, respectively.

error rate is $2\delta$ even when the average error rate is $\delta(1 + 2\epsilon)$. In addition, unlike the composition of classifiers in Example 1, the margin of which is $2\epsilon$, the margin of the new composition of classifiers is $1-2\delta(1+2\epsilon)$, which can be any value smaller than 1. Taking $\epsilon \rightarrow 0$ concludes that the first-order upper bound (1) is also tight when the margin is arbitrarily high. A visual illustration of the composition of classifiers is given in Figure 12b.