# OpenReview forum: "When are ensembles really effective?"
_NeurIPS.cc/2023/Conference — NeurIPS 2023 poster_

### Official Review · Reviewer_E93x · 2023-06-26

**Soundness:** 4 excellent
**Presentation:** 4 excellent
**Contribution:** 4 excellent
**Rating:** 7
**Confidence:** 4

**Summary:**

The authors offer a novel theoretical analysis of the majority vote classifier. In the process they define the "ensemble improvement rate" and give lower and upper bounds on it in terms of another entity they call the "disagreement-error ratio", under a novel assumption (competence) that they show empirically is mild. They also show how other works can be phrased in terms of lower/upper bounds on the ensemble improvement rate.

**Strengths:**

As they say in the paper, without any assumptions all that can be said is that EIR (the ensemble improvement rate) is >=-1. In MLIS20 they make additional assumptions and yield better results, but the current work improves upon the work in MLIS20 by a factor of 2.

The main novelty and strength of the paper is in the definition of "competence", a seemingly mild assumption that allows to get much better theoretical results. (Theorem 1: competent ensembles never heart, i.e. EIR>=0; and Theorem 2 giving explicit bounds of EIR under the assumption of competence.)

Another strength of the paper is that they are not satisfied that "competence" feels like a mild assumption, they show empirical proof that this is often the case.

The mathematical definitions are elegant, and the paper is easy to read.

**Weaknesses:**

It is not clear how much the analysis can translate to regression tasks, perhaps some words to address this would be helpful. It may also be nice to say some words about whether the bounds become trivial as K goes to infinity.

**Questions:**

This is a bit nit-picky, but I find the $W_{\rho}$ notation confusing: $E_{h\sim\rho}(1_{(h(X)\neq Y)})$ should really be $E_{h\sim \rho}(1_{(h(X)\neq Y)}|X, Y)$. On the other hand, others may find the notation "$E_{h\sim\rho}(1_{(h(X)\neq Y)}|X, Y)$" confusing, despite being exactly correct. (Conditional expectation on random variables yields a random variable.)

Perhaps the solution is to keep $W_{\rho}$ as $W_{\rho}(X,Y)$ (meaning not to neglect $X$ and $Y$ in the notation)? Or maybe explain a little more that $P(W_{\rho} \in [t, 1/2))$ is over the distribution of $(X,Y)$?

I am also a little confused about the relationship between competence and a similar notion where you change the order of expectation: replace $P_{(X,Y)\sim D}(E_{h\sim \rho}(1_{(h(X)\neq Y)}|X, Y) \text{is in some interval})$ with $P_{h\sim \rho}(E_{(X,Y)\sim D}(1_{(h(X)\neq Y)}|h) \text{is in some interval})$. Is this more or less intuitive? Can anything be said about ensembles with this alternate definition of competency?

How does this work compare with the seminal "Super Learner" paper (Van der Laan, M. J., Polley, E. C., and Hubbard, A. E.) that is commonly cited in theoretical ensemble research? They compare ensemble performance to the best performing base learner, but they don't give guarantees for when it is better. Would they be able to benefit from a regression-equivalent of "competence"?

**Limitations:**

None that I can see.

---

> ### Author Rebuttal · Authors · 2023-08-09
>
> We thank the reviewer for their positive review and very insightful questions. We completely agree on the notational points, and will improve this in an updated draft of the paper.
>
> As we mention in our response to Reviewer 3, the analysis is actually very simple in the regression case. In this case, via a bias-variance decomposition, we obtain an _identity_ $EIR = \frac{E_f[L(f)] - L(\bar{f})}{E_f[L(f)]} = \frac{E_X[\text{Var}_{f}[f(X)]]}{E_f[L(f)]} \geq 0$. Much of the novelty of this work is in addressing the much more challenging case of classification with the 0-1 loss. We will provide some additional clarification on this point in an updated draft of the paper.
>
> The case $K\to \infty$ is actually very interesting and highlights a significant strength of our bound versus prior work. Note that in the limit of large $K$, Theorem 2 reduces to $\text{EIR} \geq 2* \text{DER} - 3$. Far from trivial, this is in fact the same bound one would obtain in the _binary_ case according to the result of Masegosa et al. 2020. Prior to our work, this bound was the tightest known for any $K$. So even under the pessimistic scenario $K \to \infty$, our results (and the competence assumption) constitute a significant improvement over existing second-order bounds.
>
> With respect to the alternative form of the competence assumption the reviewer proposes, we could see a case for this new notion being slightly more intuitive. In particular, depending on one's perspective, it may be more intuitive to reason about a probability over models $h\sim \rho$ versus a probability over data. However, it's unclear whether a similar analysis could be conducted under this assumption -- certainly the techniques we employ would have to be suitably adjusted.
>
> The "Super Learner" paper you mention proposes and studies a specific method for ensembling, whereas our paper derived results that apply to (almost) any arbitrary ensembling technique. Moreover, they focus on the regression setting, whereas our focus is on classification.

---

> > ### Comment · Reviewer_E93x · 2023-08-17
> >
> > I thank the authors for the interesting comments. I retain my score, and recommend acceptance.
> >
> > About "Super Learner", I think the authors are mistaken: stacked generalizations are not a "specific method for ensembling", but rather a vast generalization of almost all ensembling techniques used in practice. One would be hard pressed to find an ensembling method that does not fall into that category. (Many classic ensembling methods such as simple mean/median etc. are simply the case where the stacker is constant.) I'd still urge the authors to make the connection and explain the differences in the final draft.

---

> > > ### Author Response · Authors · 2023-08-22
> > >
> > > Thank you again for your positive feedback. It seems we may have misinterpreted the results of the “Super Learner” paper; we will do a more thorough review of this work and include a discussion in an updated draft of the paper.

---

### Official Review · Reviewer_un8p · 2023-07-05

**Soundness:** 3 good
**Presentation:** 4 excellent
**Contribution:** 4 excellent
**Rating:** 9
**Confidence:** 5

**Summary:**

The paper "When are ensembles really effective?" discusses the question asked in the title: When are ensembles really effective? To do so, the authors introduce novel theoretical measures called the ensemble improvement rate and the disagreement-error ratio. The ensemble improvement rate measures the relative rate an ensemble improves over a single learner, whereas the disagreement-error ratio measures the expected disagreement between members of the ensemble scaled by the expected performance of each model. The authors relate both terms to each other through a series of inequalities, i.e. provide an upper and lower bound for the ensemble improvement rate wrt. to the disagreement-error ratio. Moreover, the authors introduce the concept of competent ensembles which more closely resemble ensembles we are seeing in practice: While we might construct ensembles with arbitrarily bad members, we usually see ensembles that, on average, have well-performing members. Last, the authors conclude the paper with an empirical investigation that shows that a) ensembles are indeed competent in practice and b) that the disagreement-error ratio is valuable in predicting if an ensemble might improve performance compared to a single model.

**Strengths:**

- The paper is well-written and easy to follow. The authors introduce remarks where necessary and re-visit existing results from the literature if it adds valuable information to the paper.
- The authors introduce new theoretical concepts, argue in favor of them, and show their practical value in experiments. In addition, the authors also validate any assumptions they are making in their empirical investigation. Hence, the authors offer a very complete overview of their ideas, studying them from a theoretical and a practical point of view. This should be the gold standard for all papers introducing novel concepts.
- The introduced ensemble improvement rate and disagreement-error ratio seem to work well in practice and hence are valuable. Second, I think the idea of competent ensembles can be potentially impactful for future studies about ensembles, as it brings the theory about ensembles closer to what we see in practice

**Weaknesses:**

- In some form, the paper presents many things we already know in a new form. For example, the disagreement-error ratio can be seen as some form of diversity that has been studied for a long time in the literature. Similarly, the concept of weak learnability and competent ensembles seem to be closely related to each other (see my question below). While the authors did a decent job relating their work to existing work, a deeper discussion here would be interstring
- There is a minor typo in the paper: Section 3 uses "K" as the number of classes, whereas in Section 4 "C" is used for the number of classes


**Questions:**

- In the introduction, you present the following scenario: "In particular, consider the following practical scenario: a practitioner has trained a single (perhaps large and expensive) model, and would like to know whether they can expect significant gains in performance from training additional models and ensembling them" -- How can you answer this question with EIR and DER? As far as I can see, EIR and DER require already trained ensembles and explain their performance and not predict the performance of a yet untrained ensemble? In this sense, it is similar to the bias-variance decomposition that explains the performance of a model/ensemble post-hoc (i.e. after training) and, therefore might be used as a theoretical guidance tool but not necessarily as a prediction of what model/ensemble performs better or worse in a given scenario. Did I miss something here?
- Did you explore the relationship between the disagreement-error ratio and the bias-variance decomposition, and if so, what is it? I assume that given a suitable disagreement rate D, one could come up with a term that is close to the variance (i.e., diversity), but the normalization via the expected error rate this novel here.


**Limitations:**

The authors empirically validate their assumption of competent ensembles that might limit the applicability of the presented theory. Moreover, they point out the unique case of Random Forests that arguably justifies a study in its own right. I agree with the authors that there do not seem to be any negative societal impacts from the theory presented in this paper.

---

> ### Author Rebuttal · Authors · 2023-08-09
>
> Thank you for your strong review of our work, and very helpful feedback.
>
> Thank you for noticing the K/C typo -- we will correct this. We also agree that a better discussion of the relation of our work to weak learnability would be helpful; we will do our best to incorporate this in an updated draft of the paper.
>
> For the first question, what you say is indeed true with respect to our theory: to compute any of the terms in our bounds (e.g. the DER), one needs to already have multiple trained models. However, one concrete piece of guidance we can give would be based on our empirical results regarding the behavior of the DER in and out of the interpolating regime. In particular, a simple rule-of-thumb would be that if a *single* trained model is _not_ able to easily interpolate the training data, then ensembling is likely to help signifcantly (though what "significantly" means will of course depend on the particular application). In the future, it would be interesting to see if the $DER </> 1$ condition could be used to obtain different heuristics in other settings.
>
> As for the second question, our results are indeed very closely related to a bias-variance decomposition. In the regression case, when one computes a bias-variance decomposition over randomness in the sampling of models, a completely analogous result holds as an identity:
> $E_f[L(f)] = L(\bar{f}) + E_X[\text{Var}_{f}[f(X)]]$, implying that
>
> $$
> EIR = \frac{E_f[L(f)] - L(\bar{f})}{E_f[L(f)]} = \frac{E_X[\text{Var}_{f}[f(X)]]}{E_f[L(f)]}
> $$
>
> Thus, without the normalization, the bound in Theorem 2 admits a similar interpretation. Similar expressions can also be derived for other loss functions, e.g. as in Ortega et al. 2022. We agree that this is an important comment, and will use additional space in an updated draft to include a discussion on this point.

---

> > ### Comment · Reviewer_un8p · 2023-08-22
> >
> > Thank you for the clarifications, which solidified my original scoring :-)

---

### Official Review · Reviewer_sRJm · 2023-07-05

**Soundness:** 3 good
**Presentation:** 3 good
**Contribution:** 3 good
**Rating:** 7
**Confidence:** 4

**Summary:**

The paper asks the following question: when is ensembling beneficial? The benefit is measured by the ensemble improvement rate (EIR), defined as the difference of average and majority voting risk, divided by the average risk. It is shown that 'competent ensembles never hurt', in the sense that under mild assumptions, EIR \geq 0 (Th. 1). A more precise result (Th. 2) provides quantitative bounds relating EIR and another quantity called the disagreement-error ratio (DER). DER is simply the expected disagreement between our ensembles, divided by the expected loss. A consequence of this result is that improvement is moderate when disagreement is small, whenever it can be large whenever disagreement is large. The paper concludes with some empirical evaluation.

**Strengths:**

The key question of the paper is of utmost importance to the machine learning community. The answers provided by the present work, although simple, are quite interesting. What is really promising is that competence can be evaluated in practice (Section 4.2), giving an actionable way to check whether it is profitable to do ensembling in a specific concrete case.

**Weaknesses:**

The main weakness in my opinion is the restriction to the classification setting, and, further, to the use of the 0-1 loss.

**Questions:**

Do you suspect that Th. 2 is tight? Could examples similar to those developed for Th. 1 achieve tightness in the string of inequalities?

**Limitations:**

see weaknesses

---

> ### Author Rebuttal · Authors · 2023-08-09
>
> We thank the reviewer for their positive review and helpful feedback. The classification/0-1 loss setting represents the most significant challenge from the theoretical perspective, hence the focus on this case. For example, in regression with MSE loss, one has (via a simple bias-variance decomposition) the following identity:
>
> $$
> E_f[L(f)] = L(\bar{f}) + E_X[\text{Var}_{f}[f(X)]]
> $$
>
> From this, an analogous characterization of the EIR can be derived: $EIR = \frac{E_f[L(f)] - L(\bar{f})}{E_f[L(f)]} = \frac{E_X[\text{Var}_{f}[f(X)]]}{E_f[L(f)]} \geq 0$. Similarly, for other loss functions (like the cross entropy loss), the analysis is significantly easier, as one can typically exploit Jensen's inequality to relate the average error to the error of the ensemble model. Other works have addressed this case in detail, e.g. Abe et al. 2022 and Ortega et al. 2022. We will provide some additional clarification on this point in an updated draft of the paper.
>
> Without further restrictions, we believe Theorem 2 should be tight, although we do not have a formal argument for this. This belief comes from numerical simulations involving perturbations of examples along similar lines to those given in Appendix D. Consider for each example $x$, exactly $(1-\epsilon)$ fraction of classifiers predict the correct label and the remaining $\epsilon$ fraction of classifiers predict a wrong label. In this case, $EIR=1$, and the lower bound of the Theorem 2 is $1-\epsilon\frac{4(K-1)}{K}$ which can be arbitrarily close to $1$. More precise conditions on the nature of the constants in the lower bound of Theorem 2 arise from considering classifiers with slightly differing individual probabilities.

---

> > ### Comment · Reviewer_sRJm · 2023-08-22
> > **Post-rebuttal**
> >
> > Thank you for the added clarification regarding the loss functions and the comment on the tightness of Th. 2.

---

### Official Review · Reviewer_GNtk · 2023-07-06

**Soundness:** 2 fair
**Presentation:** 3 good
**Contribution:** 2 fair
**Rating:** 4
**Confidence:** 2

**Summary:**

Analysis the ensemble improvement rate vs. the disagreement-error ratio

**Strengths:**

A solid analytical analysi.

**Weaknesses:**

It is not clear for me what is the usefulness of obtained results.
The scope of the experiments is limited.

**Questions:**

What is the usefulness of this study?
How to apply them in practice?

---

> ### Author Rebuttal · Authors · 2023-08-09
>
> Thank you for providing a review of our paper. As the title and results suggest, the usefulness of this study is in describing when we can expect ensembling methods to be effective. In particular, our empirical result suggests that ensemble improvement is much less pronounced for interpolating versus more traditional noninterpolating ensembles. If possible, it would be helpful if you could provide some additional feedback and/or detail regarding your concerns about our work.

---

### Official Review · Reviewer_G3j5 · 2023-07-28

**Soundness:** 4 excellent
**Presentation:** 4 excellent
**Contribution:** 3 good
**Rating:** 7
**Confidence:** 4

**Summary:**

This study explores when ensembling leads to significant performance enhancements in classification tasks. Theoretical findings reveal that ensembling improves performance considerably when the disagreement rate among classifiers is large compared to the average error rate. The study also establishes improved upper and lower bounds on the average test error rate of the majority vote classifier under a condition called competence. In addition to the theoretical analysis, the researchers conduct empirical experiments in various scenarios to validate their theory. The study highlights a distinction in the behaviour of interpolating models  and non-interpolating models, showing that ensembling provides more significant benefits in the latter case.


**Strengths:**

- The work addresses a relevant and, still, open problem: when are ensembles really effective? The presented theoretical framework is specially designed to answer this question. Although it builds on previous theoretical analysis, it introduces a new concept, “competent ensembles”, which give rises to more insightful conclusions.

- The empirical evaluation of this work is quite strong. Besides validating the proposed theoretical framework, authors also provide insightful analysis about the effectiveness of ensembles. The analysis of what happens at the interpolation regime is of high importance for current practices of machine learning.


**Weaknesses:**

- Assumption 1 is central in the theoretical analysis of this work. Although authors make an effort in describing assumption 1 as a condition guaranteeing that ensembles are better than individual classifiers, this assumption is very technical and it is not well described.

- Although this is not the main aim of this paper, the provided theoretical framework provides little to no guidance in how to build better ensembles.


**Questions:**

Could you derive Thm 2 under the assumption that EIR>1? Because, in this case, your assumption should be EIR>1.


**Limitations:**

Authors do not fully discussed the limitations of their work.

---

> ### Author Rebuttal · Authors · 2023-08-09
>
> We thank the reviewer for their positive review of our work, and useful feedback. While the aim of this work is not necessarily to provide guidance on how to build better ensembles, we hypothesize that some of our insights might lead to the development of better methods in future work. In particular, our findings regarding EIR/DER in and out of the interpolating regime suggest that overparameterized deep ensembles require more diversity than current methods allow to be effective.
>
> Regarding Assumption 1, there are a few comments that can be made to perhaps better elucidate its nature. For example, taking $t = 0$ in the statement of the assumption reveals the condition that, on average over the data, more classifiers in the ensemble are correct than not. What we actually require is stronger than this -- that, over the data, it is more likely that *slightly* more classifiers in the ensemble are correct than *slightly* less. This is along the lines of a stochastic dominance condition. However, we do observe that the competence condition holds widely in practice. We will add these comments in the revised version of the paper.
>
> With regard to the reviewer's question about $\text{EIR} > 1$, some clarification would be helpful. By definition, $\text{EIR} \leq 1$, so perhaps the reviewer had in mind a different scenario?

---

> > ### Comment · Reviewer_G3j5 · 2023-08-14
> >
> > Dear authors,
> >
> > In relation to my question. Sorry, it was a typo. My question is: Could you derive Thm 2 under the assumption that EIR>0? Because, in this case, your assumption should be EIR>0. I would like to see your response.

---

> > > ### Author Response · Authors · 2023-08-16
> > >
> > > Thank you for clarifying, this is a good question. It turns out that assuming $EIR > 0$ is _not_ sufficient to prove Theorem 2 (that is, competence is not necessary for $EIR \geq 0$). To see this, we can provide a numerical example for when $EIR > 0$ but the ensemble is not competent, and the lower bound in Theorem 2 does not hold. We summarize the setup of this example in the following table:
> > >
> > > | Set $A \subset \mathcal{X}\times \mathcal{Y}$, $P_{X,Y}(A) = 0.6$           | Set $B\subset  \mathcal{X}\times \mathcal{Y}$, $P_{X,Y}(B)=0.4$        |
> > > |------------------------|---------------------|
> > > | For all $(x,y)\in A$, $P_h(h(x) = y) = 0.7$ | For all $(x,y)\in B$, $P_h(h(x)=y) = 0.4$ |
> > >
> > > where here we assume we are in the binary $K=2$ case. For this setup, one can compute $L(h_{MV}) = 0.4$ (note $h_{MV}(x) \neq y \Leftrightarrow (x,y)\in B$), $E[L(h)] = 0.42$, and $E[D(h,h')] = 0.444$. In this case, $EIR = 0.02/0.42 > 0$, but the lower bound is $DER - 1 = 0.444/0.42 - 1 = 0.024/0.42 > 0.02/0.42 = EIR$, and so the lower bound in Theorem 2 does not hold.

---

### Decision · Program_Chairs · 2023-09-21

**Decision:**

Accept (poster)

**Comment:**

This paper theoretically explores when ensembling leads to significant performance enhancements in classification tasks. Reviewers were unanimously convinced by the contribution and their positive assessment was reinforced by the authors' response (with the exception of one reviewer, who was not very confident in their recommendation and did not engage into the discussions). I recommend acceptance.